# Orthopedic Surgeons’ Accuracy When Orienting an Acetabular Cup. A Comparison with Untrained Individuals

**DOI:** 10.3390/medicina58070973

**Published:** 2022-07-21

**Authors:** Jesús Moreta, Óscar Gayoso, Daniel Donaire-Hoyas, Jorge Roces-García, Jesús Gómez-Vallejo, Esther Moya-Gómez, David Raya-Roldán, Alberto Albert-Ullibarri, Fernando Marqués-López, Jorge Albareda

**Affiliations:** 1Department of Orthopedic Surgery and Traumatology, Hospital Galdakao-Usansolo, 48960 Galdakao, Spain; 2Department of Orthopedic Surgery and Traumatology, Hospital San Rafael, 15006 A Coruña, Spain; ogayoso@yahoo.com; 3Department of Orthopedic Surgery and Traumatology, Hospital de Poniente, 04700 El Ejido, Spain; ddhmp5@hotmail.com (D.D.-H.); david9323@hotmail.com (D.R.-R.); albertullibarri@gmail.com (A.A.-U.); 4Department of Construction and Manufacturing Engineering, Polytechnic School of Engineering of Gijón, University of Oviedo, 33204 Gijón, Spain; rocesjorge@uniovi.es; 5Department of Orthopedic Surgery and Traumatology, Hospital Clínico Universitario Lozano Blesa, 50009 Zaragoza, Spain; jgomezvallejo@yahoo.es (J.G.-V.); albaredajorge@gmail.com (J.A.); 6Department of Orthopedic Surgery and Traumatology, Hospital de la Santa Cruz y San Pablo, 08025 Barcelona, Spain; emoya@santpau.cat; 7Department of Orthopedic Surgery and Traumatology, Parc de Salut Mar, 08003 Barcelona, Spain; fmarques@psmar.cat

**Keywords:** accuracy, acetabulum, arthroplasty, hip prosthesis, navigation, total hip replacement

## Abstract

*Background and Objectives:* Previous studies demonstrated a huge variability among surgeons when it comes to reproducing the position of an acetabular cup in total hip arthroplasty. Our main objective is to determine if orthopedic surgeons can replicate a given orientation on a pelvic model better than untrained individuals. Our secondary objective is to determine if experience has any influence on their ability for this task. *Materials and Methods:* A group of specialist orthopedic hip surgeons and a group of volunteers with no medical training were asked to reproduce three given (randomly generated) acetabular cup orientations (inclination and anteversion) on a pelvic model. Error was measured by means of a hip navigation system and comparisons between groups were made using the appropriate statistical methods. *Results:* The study included 107 individuals, 36 orthopedic surgeons and 71 untrained volunteers. The mean error among surgeons was slightly greater as regards both inclination (7.84 ± 5.53 vs. 6.70 ± 4.03) and anteversion (5.85 ± 4.52 vs. 5.48 ± 3.44), although statistical significance was not reached (*p* = 0.226 and *p* = 0.639, respectively). Similarly, although surgeons with more than 100 procedures a year obtained better results than those with less surgical experience (8.01 vs. 7.67 degrees of error in inclination and 5.83 vs. 5.87 in anteversion), this difference was not statistically significant, either (*p* = 0.852 and *p* = 0.981). *Conclusions:* No differences were found in the average error made by orthopedic surgeons and untrained individuals. Furthermore, the surgeons’ cup orientation accuracy was not seen to improve significantly with experience.

## 1. Introduction

The clinical results obtained following total hip replacement (THR) are very much dependent on the way components are oriented. The orientation of the acetabular component influences range of motion [1,2,3], the risk of dislocation [4,5,6,7,8,9], the wear rate [6,8,10,11,12], the functional result [13,14], the incidence of squeaking [15,16,17] and implant survival [18]. However, what constitutes optimal orientation is still a matter of controversy [3,5,7,8,19]. Additionally, the wide range of anatomical, radiological and surgical definitions proposed in an attempt to establish an ideal pattern have done little to bring clarity to the situation [20,21,22].

Providing an appropriate definition of optimal inclination and anteversion is, however, only part of the problem. Indeed, reproducing the optimal standard in the operating room can be challenging, with several studies highlighting the difficulties inherent in obtaining consistent cup positioning results [13,19,23,24,25]. Some surgeons endeavor to reproduce the recommended standard by following anatomical landmarks [26,27,28,29,30,31,32], while others strive to achieve specific angulations. Either way, it seems clear that the surgeon’s technical performance plays a decisive role in the results obtained [33].

Previous studies documented the huge variability among surgeons when it comes to determining angles or reproducing the position of an acetabular cup [34,35,36], with no significant differences being found between the results of residents and those of experienced surgeons. Graham et al. [36] detected inaccuracies in visual angle determination after analyzing the proficiency of surgeons at estimating the angles formed by two Kirschner wires fixed to bone models. Duren et al. [35] reported an unacceptably high mean error when they asked orthopedic surgeons to orient a cup to an incliniation of 40 degrees on a pelvic model, and similar results were published by Grammatopoulos et al. [34].

It must be noted, however, that none of these studies used a control group to carry out a comparative evaluation. For that reason, based on our working hypothesis that orthopedic surgeons are unlikely to obtain a more precise positioning of the acetabular cup on a pelvic model than the general population, this study will establish a comparison between the results obtained by experienced surgeons and those of a sample of individuals with no previous training. The secondary goal of the study will be to determine whether the ability to successfully orient an acetabular cup significantly evolves with increasing surgeon experience.

## 2. Materials and Methods

### 2.1. Study Design and Participants

This was an in vitro study comparing a group of orthopedic surgeons specializing in hip surgery with a group of volunteers with no medical training (control group). All participants gave their informed consent to participate in the study. The workflow is summarized in Figure 1.

Orthopedic surgeons (*n* = 36): The subjects from this group were recruited among the participants in the 2021 National Congress of the Spanish Hip Society (SECCA). They were all specialist hip surgeons or residents in training. To facilitate a segmented analysis, participants were categorized according to their professional status (resident, specialist or head of department) and experience (number of hip surgeries performed in one year).Volunteers with no medical training (*n* = 71): Subjects were recruited among the staff of a medical device distributing company (MBA SURGICAL EMPOWERMENT, Gijón, Spain), as performed by Silberberg et al. [37]. Anybody who, on account of their position, had previous training in the field of hip surgery was excluded from participating in the study.

**Figure 1 medicina-58-00973-f001:**
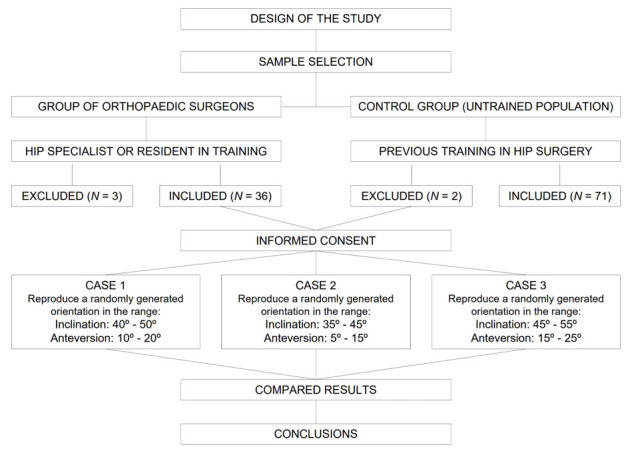
Workflow of the study.

### 2.2. Measurement System

Cup orientation was measured using the NaviSwiss hip navigation system (NaviSwiss AG, Brugg, Switzerland), a portable device equipped with an accelerometer that provides real-time measurements (inclination, anteversion, limb length and offset) without the need of previous images. The system is based on a series of tracking tags attached to the pelvis and the femur, which provide the desired measurements regardless of the way the patient is positioned. Its accuracy is ±0.3 for measuring angles and ±0.2 for measuring distances.

### 2.3. Anatomical Model and Set-Up

A plastic anatomical model was used, made up of a pelvis and a simulated cup impactor. Once the model was mounted on a flat table, initial calibration of the navigator was performed. The model was not fastened to the table so that each subject could orient the model in the direction they felt most comfortable with. (Figure 2).

### 2.4. Procedure and Tests Carried Out

Before each experiment, each subject was informed what tests they would have to carry out. An explanation was provided to the members of the control group of the concepts of inclination (or abduction angle) and anteversion, and of how they were to be measured on the pelvic model used in the experiment. Participants were not allowed to try the navigator before the actual experiment; nor were they shown their results until completion of the experiment to avoid bias in the interpretation of results. 

All subjects were asked to replicate orientations that were randomly generated by a computer within intervals that were different for each attempt:Case 1: inclination between 40 and 50 degrees, anteversion between 10 and 20 degrees.Case 2: inclination between 35 and 45 degrees; anteversion between 5 and 15 degrees.Case 3: inclination between 45 and 55 degrees; anteversion between 15 and 25 degrees.

Moreover, each subject performed each required test without being informed about the result of the previous one to prevent them from learning from previous attempts. Time was unrestricted on each attempt.

### 2.5. Measurement Error

Measurement error was defined as the difference, in absolute terms, between the inclination or anteversion obtained at each attempt and the value proposed. Absolute values were used to avoid error over- and under-compensation. Adoption of this methodology could, on the one hand, result in greater mean errors than those reported in the literature, and lead to lower dispersion rates on the other. 

### 2.6. Statistical Analysis and Software Used

A descriptive analysis was performed of the data using measures of central tendency and dispersion. Parametric and non-parametric tests were used to compare the differences in means as a function of the normality of the samples (as measured by means of the Kolmogorov–Smirnov test). Qualitative variables were analyzed using Pearson’s chi-squared test or Fisher’s exact test, depending on the magnitude of the values expected. In all cases, a *p* value of less than 0.05 was considered statistically significant.

Random numbers were generated using a Microsoft PowerPoint 365 (Microsoft Corp., Redmond, WA, USA) macro, while the analysis of data was carried out with IBM SPSS v.26 software (IBM Corp., Armonk, NY, USA). 

This section is divided by subheadings. It provides a concise and precise description of the experimental results, their interpretation, as well as the experimental conclusions that can be drawn.

## 3. Results

### 3.1. Description of the Sample

A total of 107 individuals participated in the study; 36 (33.6%) were included in group A (orthopedic surgeons) and 71 (66.4%) in group B (control). The surgeons’ group comprised 4 residents, 25 consultants and 7 heads of department (Table 1). Eighteen (50%) surgeons carried out more than 100 THR procedures a year.

The groups were not homogeneous regarding age (47.1 vs. 40.2 years; *p* = 0.005) or sex distribution (30 males/6 females vs. 40 males/31 females; *p* = 0.006). Table 2.

### 3.2. Results of the test: Positioning of the Acetabular Cup according to the Proposed Parameters

Participants were assigned three different cases, with given (randomly selected) inclinations and anteversions, which they had to replicate on the pelvic model. The aggregate data is summarized in Table 3 and illustrated in Figure 3.

The mean error among surgeons was slightly greater as regards both inclination (7.84 ± 5.53 vs. 6.70 ± 4.03) and anteversion (5.85 ± 4.52 vs. 5.48 ± 3.44), although statistical significance was not reached (*p* = 0.226 and *p* = 0.639, respectively).

An analysis of the evolution of the mean error at each attempt (Figure 4) shows that the smallest mean error was recorded at the first attempt, which precludes a potential cumulative learning effect.

To find out whether experience influences the subjects’ accuracy in replicating angles, participants were divided into those who performed over 100 THRs a year and those who did not reach that figure. An analysis of the results (Table 4) shows that, although seasoned surgeons were more likely to better determine cup inclination (8.01 vs. 7.67 degrees of error), differences were not statistically significant (*p* = 0.852); nor were there any statistically significant differences found when analyzing the influence of experience on determining cup anteversion (5.83 vs. 5.87; *p* = 0.981).

In order to find out whether professional status has an influence on the results obtained, participants were divided into residents, consultants and heads of department. This comparison did reveal a somewhat more accurate performance of more senior surgeons regarding inclination (9 degrees of error for residents; 7.84 for consultants; and 7.19 for heads of department) and anteversion (6.67 degrees vs. 5.93 and 5.10, respectively). However, these differences were not statistically significant, either (*p* = 0.879 and *p* = 0.854, respectively).

As regards error types, surgeons were found to underestimate inclination (*p* < 0.001) as 77.8% of their attempts resulted in placement of the cup in a more-horizontal-than-intended position. However, the errors made by members of the control group did not follow any trend (45.1% were placed more horizontally than required and 51.2% more vertically than required). Similarly, surgeons tended to err towards less anteversion than intended (69.4%), while this was not the case among the general population (42.7% erred towards less anteversion and 51.2% towards more anteversion) (*p* < 0.001) as seen in Table 5.

Given that significant differences were found between the groups with respect to the magnitude of the errors made, an analysis was made of the aggregate data to see whether other variables (such as sex or age) could affect the subjects’ accuracy. Both sexes obtained equal results for inclination (7.09 degrees de error for men vs. 7.08 degrees for women), although men performed better as far as anteversion was concerned (5.36 degrees vs. 6.08); this difference was not, however, statistically significant (*p* = 0.353). As regards age, no correlation was found between this parameter and inclination errors (ρ = 0.065) or anteversion errors (ρ = 0.022).

## 4. Discussion

Two kinds of errors have been identified in motor learning. Constant error measures deviation from the target, with missed attempts tightly clustered around one another. Variable error refers to scattered misses around a target without a trend or pattern being present [38]. The study of motor learning has shown that the measure of error that is most sensitive to the effects of repetitive practice is consistency, i.e., a reduction in random error [39].

The most striking finding of this study is that the orthopedic surgeons analyzed were not better at determining angulations in space than the subjects in the control group. The magnitude of the errors observed, and their high variability, could be due to a lack of specific training. Furthermore, their level of performance of the task did not appear to improve with years of experience or to depend on their position in the hierarchical ladder.

The evidence suggests that although a ±10-degree variability in cup placement is generally considered acceptable [5], mean variability stands at ±15 degrees, even among expert surgeons [19,33]. According to more recent evidence, ±15 degrees is enough to reduce the dislocation rate but ±5 degrees is necessary to optimize clinical results [13]. The mean error found in the surgeons’ group was considerably higher, with mean values of 7.84 degrees (±5.53) for inclination and 5.85 degrees (±4.52) for anteversion.

These figures do not mean that the general population is as skilled as an orthopedic surgeon at properly orienting a cup. Proper cup orientation requires a previous planning phase, in which the patient’s characteristics and the type of approach to be conducted should be factored in, as well as a thorough knowledge of human anatomy to be able to find the required landmarks [26,27,28,29,30,31,32]^,^ and an understanding of the general biomechanics of the hip joint and its relationship with the spine [40]. If all this is compounded with good spatial orientation, results are likely to be satisfactory and surgeons will be able to more effectively harness their knowledge [33]. Spatial orientation training is therefore an undeniable boon for any orthopedic surgeon. However, given that organizing such training could be challenging in certain contexts, equipping operating rooms with a navigation system that supplements the surgeon’s skills may constitute a valid alternative.

There is a wide array of tools that can be used to optimize cup orientation intraoperatively. These include mechanical alignment guides, computer-assisted navigation systems, robots and accelerometers [35]. Their respective advantages and drawbacks have been extensively discussed in the literature [20,41,42,43], but do not fall within the scope of this article.

Inaccuracies in visual angle determination were already reported by Graham et al. [36], who analyzed the proficiency of 31 surgeons at estimating the angles formed by two Kirschner wires fixed to bone models. The mean error during visual determination was of 5.4 degrees (±5.3), which fell to 0.8 degrees (±0.9) when using a smartphone-based inclinometer. Duren et al. [35] found similar results, reporting a mean error of 6.4 degrees (±4.4) when they asked 18 surgeons to orient a cup to an inclination of 40 degrees on a pelvic model. The magnitude of the error decreased to 1.7 degrees (±1) when the surgeons were allowed to use a digital inclinometer. In addition, while all attempts made with a freehand technique fell outside the safety zone defined in the experiment, inclinometer-assisted attempts did fall within the safety zone. In a similar experiment, Grammatopoulos et al. [34] found a mean anteversion error of 5 (±15) degrees and a mean inclination error of 5 (±10) degrees. All these in vitro studies found similar error levels with high dispersion rates (Table 6).

Although statistical significance was not reached (*p* = 0.099), the mean error among surgeons when evaluating inclination (7.84 ± 5.53) was somewhat greater than when evaluating anteversion (5.85 ± 4.52). Nonetheless, it must not be forgotten that the safety range assigned by most authors to cup inclination is greater than for anteversion, which means that the same magnitude of error may place a cup outside the anteversion safety zone but not outside the inclination safety zone. There is, however, no consensus in the literature as to whether the outcome (or survival) of the prosthesis is more sensitive to changes in anteversion or inclination [2,17,44,45,46].

Statistically significant differences were nevertheless found with respect to error type (*p* < 0.001). While surgeons tended to place the cup in a more horizontal and retroverted position than planned for, the control group did not reveal a clear trend in this regard. This trend in surgeons positioning the cup more horizontally than planned was also reported by Grammatopoulos et al. [34]; but, their subjects, unlike ours, chose more anteverted positions than preoperatively envisaged. These version-related differences could be explained by the type of approach used by each orthopedic surgeon.

No clinically or statistically significant differences were found when comparing surgeons with different levels of experience. Differences in judging inclination between surgeons carrying out more or less than 100 THRs a year was 0.34 degrees; the difference when judging angulation was 0.04 degrees. Comparing the subjects’ performance in terms of their professional status yielded similar results. Although accuracy did tend to increase in surgeons holding more senior positions, differences were neither statistically nor clinically significant; the total improvement being limited to 1.81 degrees for inclination and 1.57 degrees for anteversion. Grammatopoulos et al. [34] did not find significant differences between surgeons with varying levels of experience either. Moreover, although Duren et al. [35] do mention a trend toward fewer errors among the more seasoned surgeons, they do not offer any statistically significant results in this regard.

To understand this, it is essential to consider the nature of the task under analysis. In the first instance, although THR is a regularly performed operation, the number of procedures carried out by each surgeon is probably not large enough to result in an automated execution (as compared, for example, with most sports, where the huge number of repetitions performed are typically enough for players to develop a series of automatic behaviors). Secondly, surgeons do not normally receive feedback about the result of their maneuvers (or if they do, they receive it long after the fact). Motor learning theory defines the term “knowledge of result” as the information subjects receive about the result of their action once its execution has been completed [39]. According to this theory, there are some kinds of actions for which knowledge of result is indispensable. These include aiming tasks where subjects cannot see the target, or those where, although subjects do have all the information required, they are unable to determine whether the response is the correct one [47]. This definition is clearly applicable to the tasks subjects had to carry out in the present study.

In the light of these results, one could wonder whether training in the abilities analyzed in this study should be more actively promoted, or whether performance of such abilities should be intraoperatively supplemented by a navigation system. Logishetty et al. [48] looked into the first possibility by training two groups of students in orienting an acetabular cup on a pelvic model. One of the groups was trained by an experienced orthopedic surgeon while the other was trained using augmented reality glasses. The results of both groups were comparable and practically all participants stated that it would have been ideal to combine both modes of instruction. Gofton et al. [49] reported better results in students who had been trained using a navigator than in those who had received conventional training. In a metanalysis, Snijders et al. [50] reported that all analyzed studies found statistically higher cup placement accuracy levels (regarding both inclination and anteversion) with a navigation system than with manual techniques. In fact, Grammatopoulos et al. [34] advocated for the routine use of alignment guides as their study showed them to reduce variability in cup anteversion by one third. However, even when such guides were used, variability still stood at ±10 degrees. The present study did not include a test whereby subjects could aid themselves with the NaviSwiss Navigation System as, given that the system provides instant feedback on the way the cup is oriented, the result would have been 100% accurate for both parameters in all instances.

It seems certain that the use of more precise navigation systems is making it increasingly necessary to reach a consensus regarding what should be considered optimal cup orientation [21,51]. Moreover, once such values are defined, orthopedic surgeons ought to receive the required training or be given the necessary tools to reproduce them intraoperatively. Twenty-nine percent of the subjects analyzed by Grammatopoulos et al. [34] were not able to replicate on the pelvic model the ideal orientation that they had planned for. Use of an alignment guide made it possible to reduce this percentage to 5%, which seems to confirm that the use of that kind of tool could reduce the degree of variability.

After a meticulous search of the main scientific databases, we can assure that this is the first study to compare the spatial perception skills of surgeons with those of a control group made up of subjects with no previous medical training. Additionally, our sample size is larger than that of Graham [36], Duren [35] and Grammatopoulos [34]. Both the magnitude and the dispersion of the errors observed are indicative of significant gaps in the training received by surgeons in this area, as well as of a natural human limitation, which should be mitigated with the relevant support tools. Studies analyzing the orientation achieved with real patients have reported malalignment rates of up to 70% [23,25]. Nonetheless, the steps to be taken to address this problem extend beyond the scope of this article. Some authors advocate for supplementing the surgeons’ knowledge with technical aids that provide instant feedback [48,49]. Others believe that the solution should be based on the use of support tools during surgery [20,24,35,36,50,52,53,54], while others claim that both approaches should be combined [34]. Furthermore, it should be stated that although the present article discusses ways of improving cup orientation, there is no evidence that such an improvement will result in desirable clinical effects, such as improved results, a reduction in the complications rate, or an increase in prosthetic survival [42,50].

Our study is not exempt from limitations. The first of such limitations is that the goal of the analysis was circumscribed to evaluating the subjects’ ability to replicate a given orientation without any external aid, without consideration of other fundamental variables, such as the ability to utilize anatomical landmarks [26,27,28,29,30,31,32], or the influence of patient positioning [20,40]. Secondly, all measurements were made on a pelvic model rather than on a real patient or an anatomical specimen. Although this does not reproduce the real-world situation of an operating room, it does reduce the variability resulting from such factors as individual anatomical differences, poor visibility due to bleeding, presence of osteophytes or patient positioning [21]. Finally, the sample size—although larger than that of other studies—was relatively small [34,35,36]^,^ and the selection was made only among surgeons specializing in THR. We also do not know if the help of the usual anatomical references can help surgeons to reproduce a given orientation. Thus, a possible future orientation would be to replicate this study in anatomical specimens, in order to answer that question.

## 5. Conclusions

Errors made by orthopedic surgeons when placing an acetabular cup according to predetermined parameters tend to present with wide dispersion rates, comparable to those associated with the errors made by subjects in the control group, all of whom lacked medical training. Additionally, the surgeons’ cup orientation accuracy was not seen to improve significantly with experience or as they climbed the professional ladder. For that reason, it would seem appropriate to upgrade the training programs for hip surgeons and/or help them improve their performance by equipping operating rooms with the required support systems.

## Figures and Tables

**Figure 2 medicina-58-00973-f002:**
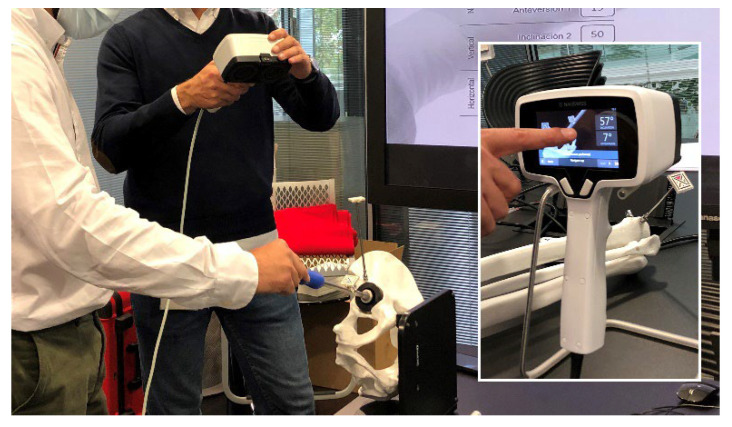
One of the volunteers performing the test. In the square, the navigator providing information on inclination and anteversion.

**Figure 3 medicina-58-00973-f003:**
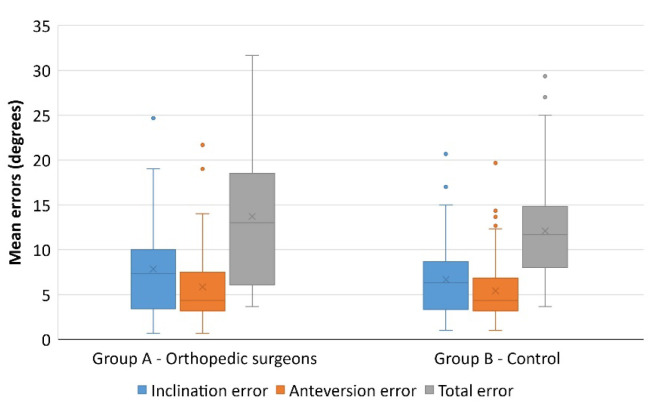
One of the volunteers performing the test. In the square, the navigator providing information on inclination and anteversion.

**Figure 4 medicina-58-00973-f004:**
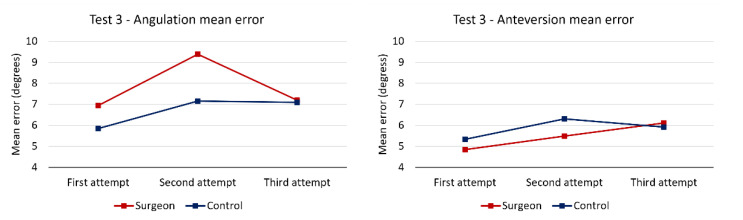
Mean angulation and anteversion errors in test 3.

**Table 1 medicina-58-00973-t001:** Description of the groups.

	*N*	%
**Group A—Orthopedic Surgeons**	**36**	**33.6%**
* Experience*		
Resident	4	11.1%
0–25 total hip replacements a year	3	8.3%
25–50 total hip replacements a year	2	5.6%
50–100 total hip replacements a year	9	25.0%
More than 100 total hip replacements a year	18	50.0%
* Position*		
Resident	4	11.1%
Consultant	25	69.4%
Head of department	7	19.4%
**Group B—Control**	**71**	**66.4%**
* Occupation*		
Clerical work	4	5.6%
Warehousing	14	19.7%
Customer service	11	15.5%
Accounts	11	15.5%
Information technology	8	11.3%
Marketing	8	11.3%
Logistics	8	11.3%
Other	7	9.9%
**Total**	**107**	**100%**

Bold-There are two groups and they were divided using different parameters (Experience and position in group A, and Occupation in group B).

**Table 2 medicina-58-00973-t002:** Age and sex distribution.

	Group A Surgeons	Group B Control	*p*-Value
**Mean age**	47.14	40.23	0.005 *
**Sex**			
Males	30 (83.3%)	40 (56.3%)	0.006 *
Females	6 (16.7%)	31 (43.7%)	

* Statistically significant differences were found.

**Table 3 medicina-58-00973-t003:** Mean errors (degrees) in test 3.

	Group A Surgeons	Group B Control	*p*-Value
Mean error—Inclination	7.84 (±5.53)	6.70 (±4.03)	0.226
Mean error—Anteversion	5.85 (±4.52)	5.48 (±3.44)	0.639
Total mean error—Inclination + anteversion	13.69 (±7.76)	12.18 (±5.32)	0.298

Statistically significant differences were found.

**Table 4 medicina-58-00973-t004:** Mean errors according to position and experience.

		Inclination Error		Anteversion Error	
	*N*	Mean	*p* Value	Mean	*p* Value
**Group A—Orthopedic surgeons**					
*Experience-based classification* *(number of surgeries a year)*					
Less than 100	18	8.01	0.852	5.83	0.981
More than 100	18	7.67		5.87	
*Seniority-based classification*					
Resident	4	9.00	0.879	6.67	0.854
Consultant	25	7.84		5.93	
Head of department	7	7.19		5.10	

Statistically significant differences were found.

**Table 5 medicina-58-00973-t005:** Mean errors according to position and experience.

	Inclination Error						Anteversion Error					
	Too Horizontal		Right Inclination		Too Vertical		Less Anteversion		Right Anteversion		More Anteversion	
	*N*	%	*N*	%	*N*	%	*N*	%	*N*	%	*N*	%
Surgeons	84	77.8	7	6.5	17	15.7	75	69.4	6	5.6	27	25.0
Control	96	45.1	8	3.8	109	51.2	91	42.7	13	6.1	109	51.2

**Table 6 medicina-58-00973-t006:** Mean error in in vitro studies with surgeons.

		Mean Error (Degrees)		
	*N*	One Angle	Inclination	Anteversion
Graham (2013)	31	5.4 (±5.3)	-	-
Duren (2020)	18	-	6.4 (±4.4)	-
Grammatopoulos (2016)	21	-	5.0 (±10)	5.0 (±15)
Present study	36	-	7.8 (±5.5)	5.9 (±4.5)

## Data Availability

Not applicable.

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
