# Peer review of "Orthopedic Surgeons’ Accuracy When Orienting an Acetabular Cup. A Comparison with Untrained Individuals"

_medicina, 2022, doi:10.3390/medicina58070973_

Round 1

Reviewer 1 Report

The authors aim to establish a comparison between the results obtained by experienced surgeons and those of a sample of individuals with no previous training for cupula position in hip arthroplasty.

- A good point of this study is that the computer assisted surgery will actually help in reducing the length of the learning curve and literature lack of studies with this kind of vision.

- The study seems methodologically good

- However, the underlying assumption of this study is that the difficulty belongs to the positioning of a cupula in a bone while, the difficulty in a surgeon practice belongs to the fact that surgeons do not see the entire pelvis and are bothered by surrounding soft tissues. Indeed, the real experience of the surgeon does not lie on his good vision but on the fact that he can represent himself the position of the pelvis that he does not see.  This implies that the global relevance of the study is highly questionable.

- There should be more information about the “non trained” population as they are from a medtech/implant company, some of them have maybe help surgeons during surgical procedure and have an experience. Even with the precision of the position of the participants in the company, this information is not given.

Abstract:

“Surgeons with more than 100 procedures a year obtained better 25 results than those with less surgical experience (8.01 vs. 7.67 degrees of error in inclination and 5.83 26 vs. 5.87 in anteversion), although the difference was not statistically significant either (p=0.852 and 27 p=0.981).” must be removed. Not significant

Methods:

Line 96: why to prefer an interval to an accurate value. It seems that it would induce biases and misinterpretation regarding the dispersion.

Line 110: there is no target value, but an interval.

Did the authors have any indication for the duration of the procedure?

Reviewer 2 Report

1.      Keywords need to be reordered based on alphabetical order.

2.      What is the novel of recent studies? There are several works have been published explaining the accuracy of acetabular cup positioning in joint replacement surgery, For example: Improving the accuracy of acetabular component orientation: avoiding malposition. Academy of Orthopaedic Surgeons. 2010. The highlighted work has been explained well in the literature. The Reviewer argues the originality of the idea and work contribution that leads to recommending rejection of the manuscript. It is an important point to highlight something really new in the manuscript.

3.      Previous research related to acetabular cup orientation with their finding and its limitation should be explained well in the introduction section to show research gap that is filled by the novelty of the present study.

4.      There are a lot of studies that do not consider acetabular cup orientation, for example, literature published in MDPI is as follows: Computational Contact Pressure Prediction of CoCrMo, SS 316L and Ti6Al4V Femoral Head against UHMWPE Acetabular Cup under Gait Cycle. J. Funct. Biomater. 2022, 13, 64. https://doi.org/10.3390/jfb13020064 that would influence the results analysis. The authors suggested including this literature in the introduction section in line 37 for supporting the functional result since it only used 1 literature. It is also can be used to explain the research gap from my previous comments.

5.      What are the criteria for sample selection? It is not explained well.

6.      It is important to explain the reference/basis/protocol for the sample used in the present study since it would be impacted the results and lead to biased results if it is mistaken.

7.      Research flow needs to be explained in a form of illustration to make the reader more interested and easier to understand rather than only dominant text with specific figures.

8.      In line 324, using the term “To the best of our knowledge…”, is not scientifical proven and not the correct way to emphasize research findings. The authors should conduct literature searching from the reputable database and present it in the manuscript to prove “….this is the first study to compare the spatial perception skills of surgeons with those of a control group made up of subjects with no previous medical training”. It is also a crucial issue that needs to addresses be the authors same as my previous comments number 2.

9.      Results comparison with previous similar literature related to acetabular cup position needs to be included.

10.   The limitation of the present research needs to be explained in the manuscript before the conclusion section.

11.   The English language is necessary to proofread due to grammatical and language style issues.

12.   Please make sure the authors have been used medicinally template properly. Comparing with published manuscript format may be done to the author for ensuring everything is appropriate. For example, is not use the number for subsection in the manuscript. Please, revise it.

13.   Enriching literature from the latest research in the last 5 years is necessary for the references. It is encouraged using literature that published by MDPI. 

Reviewer 3 Report

Discussion:

-Please, provide what is the evidence in the past and why is this study needed in comparison to the previous evidence?

- Please further discuss the details of your findings

- What are the clinical applications of your results? How they can improve the clinical practice?

- The meaning of the study: possible mechanisms and implications for clinicians or policymakers

- Strengths and weaknesses in relation to other studies, discussing particularly any differences in results

- Limitations of your study

- Unanswered questions and future research

Round 2

Reviewer 1 Report

The authors provide the expected answers and the manuscript can be published in its present form. 

Congrats to the authors

Reviewer 2 Report

I am recommend this manuscript for publication in Medicina.

This manuscript is a resubmission of an earlier submission. The following is a list of the peer review reports and author responses from that submission.